# Digital Marketing: A Unique Multidisciplinary Approach towards the Elimination of Viral Hepatitis

**DOI:** 10.3390/pathogens11060626

**Published:** 2022-05-29

**Authors:** Mohammadreza Pourkarim, Shahnaz Nayebzadeh, Seyed Moayed Alavian, Seyyed Hassan Hataminasab

**Affiliations:** 1Department of Management, Yazd Branch, Islamic Azad University, Yazd P.O. Box 89195/155, Iran; mohamadreza.pourkarim@iauyazd.ac.ir (M.P.); hataminasab@iauyazd.ac.ir (S.H.H.); 2Middle East Liver Diseases (MELD) Center, Tehran P.O. Box 14155/3651, Iran; alavian@thc.ir

**Keywords:** digital marketing, viral hepatitis, elimination, WHO, program, global, social media, influencer, campaign, management, virtual

## Abstract

New technologies are supported by the global implementation of the internet. These improvements have deeply affected various disciplines of sciences and consequently changed services such as daily business, particularly health sectors. Innovative digital marketing strategies utilize the channels of social media and retrieved user data to analyze and improve relevant services. These multidisciplinary innovations can assist specialists, physicians and researchers in diagnostic, prophylaxis and treatment issues in the health sector. Accordingly, compared to recent decades, health decision makers are more accurate and trustful in defining new strategies. Interestingly, using social media and mobile health apps in current pandemics of SARS-CoV-2 could be an important instance of the key role of these platforms at the local and global level of health policies. These digital technologies provide platforms to connect public health sectors and health politicians for communicating and spreading relevant information. Adding influencers and campaigns to this toolbox strengthens the implementation of public health programs. In 2016, the WHO adopted a global program to eliminate viral hepatitis by 2030. Recent constructive measures that have been used in the battle against COVID-19 could be adopted for the elimination of viral hepatitis program. The presented evidence in our narrative review demonstrates that the application of digital marketing tools to create campaigns on social media, armed with professional influencers, can efficiently consolidate this program. The application of different strategies in using these popular tools will raise the public awareness about viral hepatitis. Subsequently, the availability of an effective vaccine for HBV and antiviral medication for HCV can motivate the audience to take steps towards prophylaxis and screening methods against these infectious illnesses. The encouragement of health policy makers to apply digital communication technologies and comprehensive roadmaps to implement this global program will certainly decrease the burden of viral hepatitis worldwide.

## 1. Introduction

### Digital Marketing and Its Tools

The global implementation of the internet has undoubtedly supported the fast and unprecedented growth of new technologies at individual and institutional level [1]. These improvements have deeply affected various disciplines of sciences and consequently changed daily services and business. Accordingly, users of these new generation digital devices have adapted to significant alterations in daily behavior, interests, and expectations. These changes in human habits root from innovations in technologies, such as artificial intelligence (AI), block chain, virtual reality (VR) and autonomous vehicles [2]. The digital revolution in marketing has created enormous advantages as well as challenges that have led to the development of new concepts [3]. For instance, digital marketing has been recognized as one of the recent emerging transdisciplinary concepts in the world of technology. Of note, digital trends in human society have enabled individuals to communicate across boundaries of time and space, to access information around the world and apply it in marketing and business [4].

From a transdisciplinary perspective, digital marketing with the support of communicative tools can create strategic plans and forward these to a favorable destination. In other words, digital marketing has provided a series of customized platforms for communicating with specific stakeholders through channels and related tools using computers, smartphones and tablets [5]. These channels allow the gathering of information and include websites as well as different platforms of social media such as Facebook, YouTube, Twitter, Pinterest, TikTok and LinkedIn. These communication platforms provide an effective environment for interactions between the organization, stakeholders and end users. The easy accessibility, message customization, cost-effectiveness, ability to design and execute strategies and support of stakeholders are the main beneficial characteristics of this type of marketing. Therefore, marketing strategies utilize these channels and user data to analyze and improve relevant services [5]. Recent investigations show a dramatic increase in internet users, spanning all age groups, genders and ethnicities in developing countries [6]. Depending on their skill, suitability and competence in social networking, a considerable percentage of people are use more than one platform of social media [7].

Innovation in communicative technology is the main pillar of the abovementioned improvements, which has led to the development of social media [8,9]. A general characteristic of social media is to engage users and create a sense of “being and functioning virtually”. Different types of social media allow their own users not only to find and maintain communication with friends and followers, but also to observe and react to global news together with gathering and sharing this information [10,11,12].

Digital marketing through social media encourages users to engage by posting message, comments, videos or pictures [12]. In addition, viewers become reactive agents by liking, sharing, retweeting and following these posts [13]. Interestingly, digital marketing takes advantages of the interactive characteristics of social media and attempts to exert an impact on different disciplines such as human health.

In the frame of a narrative review, we tried to dive into all available scientific evidence linked to the application of digital technologies in the establishment of health programs. From both a technological and medical perspective, previous and current experiences helped us to evaluate the benefits, drawbacks, challenges, hurdles and gaps confronted with implementation of digital marketing for the viral hepatitis elimination program of WHO. Previous positive experiences with the use of digital technology on the management of health issues, particularly in the control of vaccine-preventable infectious diseases, highlight the message of this review. Here, we are addressing a series of appropriate strategies in the use of a digital toolbox to tackle a threatening public health issue. Based on our presented data, health authorities can accelerate action plans of viral hepatitis elimination and achieve the goals of the global program.

The first chapter of this review evaluates a list of digital innovations and the impacts on human health through alterations in daily behavior and interests. In this part, several examples demonstrate that data from communicative digital customized platforms have revolutionized health services (such as surveillance of pathogens or on time and appropriate actions) and stakeholders’ expectations. 

In the following part of the text, we refer to the application of social media platforms and mobile apps during the COVID-19 pandemic. Furthermore, we shed light on the misuse of these digital tools by antivax movements. In the final part, we discuss the global burden of viral hepatitis and available medical tools for controlling this public health threat. We address digital tools that have been used for COVID-19 but are still missing in the viral hepatitis elimination program. Accordingly, to fill in these existing large gaps in public awareness and to use appropriate screening, vaccination and antiviral therapy against viral hepatitis, we urge health policy makers to apply digital marketing tools such as social media. Of note, the use of these platforms and the recruitment of influencers is highly advised.

## 2. Health and Digital Technologies

### 2.1. Social Media’s Impact on Health Care System

The future of health sectors is deeply linked to the application of advanced technologies. New technologies can lead to a massive change in healthcare practice and its related business sectors in a variety of ways. Companies involved in the food and pharmaceutical industry as well as hospital and producers of laboratory equipment take advantage from new technologies to deliver their products and services to the customers and end users. Furthermore, digital innovations can assist specialists, physicians and researchers in diagnostic, prophylaxis and treatment issues [14,15]. Compared to recent decades, health decision makers are more accurate and trustful in defining new strategies. Different national and international health organizations use these social media platforms to communicate, design and disseminate their health strategies [16]. The World Health Organization (WHO) has already used social media platforms, such as Twitter and other social networks, to communicate and report on health care information [17]. This organization calls for social media to be more active in broadcasting health messages to the public, under normal circumstances and during a health crisis [18]. Based on data collected from different regions, the WHO anticipates a shortage of human resources in the health care of nearly 12.9 million worldwide by 2035. To counter the shortage of personnel, the WHO is ambitious to apply new technologies for attracting and allocating people with different specialties in healthcare [19,20]. Additionally, the management of patients has substantially improved, and personalized contact between patients and the healthcare staff is straightforward and more convincing [21]. There is an ongoing evolution in communication platforms, and those that have implement these advanced digital technologies in their health system have significantly benefited by attracting larger audiences and consumer groups (Figure 1) [22]. 

In parallel with health service providers, digital technologies are frequently used by ordinary people who search and look for answers on questions about their health on the internet [23,24]. Surprisingly, monitoring health-seeking queries on the web in different geographical regions and seasons can provide big data, which can be used to analyze real world health associated developments. For instance, it has been shown that the frequency of certain key words in the web correlate with the number of patients with influenza-like illnesses who refer to physicians in some geographical regions [25]. This sort of digital-based data analysis can support innovative surveillance systems that can assist epidemiologists in the identification of seasonal influenza epidemics [26]. The same approach has been used in the surveillance of Dengue virus [27]. Furthermore, in some tropical countries such as India, Indonesia, Singapore, Bolivia and Brazil, web search query data corresponded with the resurgence of Dengue virus [28]. Beyond pathogen surveillance, data on web behavior can be utilized as a health indicator of web users and to determine their access to healthcare facilities, including screening tests, vaccine, physicians and therapies (Figure 1) [24]. Additionally, the utility of web-based data can be beneficial for the assessment of other health issues such as contraception [29,30], diabetes [31,32], obesity [33] or cardiovascular diseases [34]. However, in the case of communicable diseases, the application of digital technology has extraordinary relevance. 

The relevance of digital innovations is highlighted when we are confronted with human pathogens that cause significant morbidity/mortality, especially during the early days of epidemics. In moments when there is lack of information and few effective therapies (e.g., vaccines and medication), a unified and appropriate response is very critical. Accordingly, it has been demonstrated that access to digitally generated data on platforms of social media can empower an immediate response or/and control the spread of viral diseases such as HIV and Ebola [24,35]. For instance, during the Ebola epidemics in Africa, a global massive web search was observed. Interestingly, a significant positive correlation was reported between social media behavior (e.g., number of tweets) and an increasing prevalence of HIV, which exemplifies the massive potential of social media’s impact on health status information of users (Figure 1) [36,37]. The analysis of such geographical location specific data retrieved from web searches helps the health authorities implement appropriate and timely actions. Via social media networking, public health researchers or clinicians have access to a wide/global audience (Figure 1) [38,39]. This allows the rapid communication of health-related information and might contribute to the improvement of the society’s health status. Several studies have shown that users follow up the advice of messages posted on social media by health advisers [6,40,41,42]. Therefore, communicating the implementation of interventions alongside messages to support healthy behavior could have tremendous value and interests. For example, research projects that offered home HIV testing through social media platforms, known as app-based intervention, has been well accepted. [41,43].

Antimicrobial resistance is another serious health concern that is highlighted in human digital communication. By 2050, 10 million deaths due to antimicrobial resistance are estimated per year, which highlights an eminent threat to human health [44]. Recent investigations have demonstrated that misunderstandings and wrong public perceptions underlie antibiotic overconsumption [45,46,47]. Furthermore, self-medication, off-label use (e.g., for curing viral infections) and suboptimal dosing of antibiotics are examples of misconception that occur in both developing and developed countries [48,49]. Several strategies, such as different assessments or educational programs, have been implemented to increase knowledge concerning antibiotic use and infectious diseases [47]. Among a variety of tools, social networking platforms such as Facebook pages, Twitter and YouTube have been efficiently used change public perception (Figure 1) [39,50,51]. The dissemination of an educational program called “antimicrobial stewardship” (ASP) on Facebook and Twitter for students of medicine is a worthy example of the application of social media technologies [50,52]. Further, to increase students’ awareness, online games about antibiotics have been launched in the UK and USA, which had a positive impact [53]. However, the use digital marketing in health system is not always a success story, as has been demonstrated in the case of measles. Despite the availability of an effective vaccine, measles is still a worldwide public health concern, with several measles outbreaks, even in developed countries, in recent years [54]. Health policy makers have used social media to promote vaccination programs; however, antivaccine movements have applied similar tools to spread disinformation [55]. More so, the activity of the antivaccination (anti-vax) movement on social media seems much higher and more successful compared to the pro-vaccines movement, which contributes to the increased vaccine hesitancy. [56,57]. A survey showed that activities of the anti-vax movement on Twitter positively correlated with the incidence of measles outbreaks and a drop in vaccine coverage [58]. Antivaccine discourses on social media majorly highlight potential vaccine side effects and deaths. Additionally, they try to link the deaths to conspiracy theories and attempt to minimize the effectiveness of vaccines. However, none of them use scientific arguments to support their claims [58]. Contrasting the anti-vax movement, communication that reverberate on social media has been able to transmit well-informed information about measles infection sequela and has assisted parents in making the correct decision for vaccination [59]. Vaccination against influenza and HPV are experiencing the same scenarios on social media in which misinformation is propagated by antivaccine movements [60,61,62].

Besides monodirectional communication and the dispatch of health messages, social media platforms enable the interactions between stakeholders, such as healthy individuals and patients, patients and specialists and ordinary people who are interested in a healthy lifestyle [63]. Nowadays, patients can communicate through social media with health specialists rather than planning visits and speaking in person [64]. This can decrease the number of hospital visits [64]. In addition to social networking sites, different applications and algorithms offered by authorities of digital markets can be used as blueprints for this bidirectional interaction [65]. The term Mobile Health (mHealth) implies the use of different applications (apps) on mobile phones or tablets, which assist in monitoring the personal health of users [66]. In such platforms, advice on daily physical or mental behavior, such as diets, activities and treatments, are suggested to the users. Eventually, the daily functioning of the user is monitored, and personalized reports are delivered [67,68,69,70,71]. 

### 2.2. Current Pandemic and Digital Revolution

From the first days of the SARS-CoV-2 pandemic, which started in December 2019 in China, digital marketing tools (e.g., social media) demonstrated their potential use in times of crisis [72,73]. Not surprisingly, the first diagnostic report of a suspected pneumonia case with unknown etiology was posted on WeChat by Dr. Li Wenliang, who later died from the same illness. Although the use of social media to disseminate information was adopted in previous epidemics such as Ebola outbreaks [74], Zika virus [75], Influenza [76], Dengue [77] and MERS-CoV [78], the wide application of digital technology in the SARS-CoV-2 pandemic was beyond previous experiences. The World Health Organization (WHO) noted that “the coronavirus disease 2019 (COVID-19) is the first pandemic in history in which technology and social media are being used on a massive scale.” [79]. Accordingly, by applying social media, health information spread quickly and was instantly shared with the public to inform people about the epidemic, prophylactic measures and treatments [80,81,82]. Furthermore, through different channels of social media, bidirectional communication was implemented during phases of lockdown for people with health-related questions and/or to increase their own awareness [82,83]. This demand was partially linked to the current COVID-19 pandemic, and other needs were related to other illnesses in which patients needed remote care, relevant advice, and medical services (Figure 1) [84,85]. 

Access to trustable information is very important, and the spread of fake news has been a major threat for the credibility of social media during the COVID-19 pandemic [86]. In the first months of 2020, plenty of disinformation was shared social media that included conspiracy theories, in which bioweapons, the involvement of Bill Gates and the implementation the 5G network were introduced as the main causes or catalyzers of COVID-19 spread [87,88]. In contrast, following the approval of SARS-CoV-2 vaccines, social media was a crucial platform for the roll-out of vaccines. However, similar to previous vaccination campaigns, individuals used social media channels to spread conspiracy theories and antivaccination disinformation to disturb the immunization program and avoid vaccinations [89,90,91]. Unfortunately, the view rate of vaccine-opposing posts related to SARS-CoV-2 was much higher than the views of pro-vaccine messages [92,93,94,95]. Besides vaccine inequity, vaccine hesitancy, which counters public health messages, is now considered a real hurdle for vaccination against SARS-CoV-2 [96,97].

The current pandemic has been a trigger for users to innovate digital marketing. These initiatives are taken by users to intensify and amplify their activities. Often, people utilize more than one platform of social media. It allows them to disseminate information from one platform to another. This type of activity, which is called cross-platform use [79,98], is often applied by the antivaccine movement to support tweets with links to YouTube. This nimble strategy tries to boost the dispersal of vaccine-opposing videos by re-tweeting antivaccine contents [99]. Unfortunately, this strategy has not been thoroughly used by health authorities so far. 

“EpiTweetr” is another innovative tool that was developed by the European Center for Diseases Control (ECDC). This tool allows epidemiologist to track possible emerging threats through searches on different platforms particularly Twitter. This tool automatically monitors, collects and processes data on Twitter and informs experts when the posted material on Twitter is not ordinary. For instance, when an increasing number of Tweets include keywords, such as the name of a pathogen, EpiTweetr automatically informs the end-user. The generated information is the result of massive data collection that has been filtered and validated by the tool. In detail, the signaling of a threat at the early stages is detected in Twitter by EpiTwittr; however, the validation of these data is necessary. Therefore, these signals are checked and approved by public health institutes and international organizations. However, to not miss potential risks, the received signals are rechecked in other platforms of social media as well.

Although EpiTweetr is a free package tool that can be used for any potential threat, the current pandemic helped the ECDC to improve this tool in terms of data collection, processing and visualization for end users. [100].

The current pandemic has highlighted the importance of mobile health apps and digital technologies on human health [101,102,103]. To combat the pandemic, huge numbers of mobile applications are available in different countries to share health information and/or trace contacts [104,105]. These applications can track the health situation of the users and prevent the dispersal of SARS-CoV-2 [106]. Eventually, these tools successfully reduced the global costs/health burden COVID-19 [107]. Apps such as mHealth have frequently been used during vaccination processes against COVID-19 and are currently used in campaigns for vaccination against HPV and influenza [108,109,110]. Undoubtedly, without digital marketing health tools, the roll-out of mass vaccinations would be more challenging. These tools have been used to contact eligible persons for vaccinations. Furthermore, individuals are regularly invited or reminded by phone message or app notifications to make an appointment and to ensure the completion of the full vaccination program (e.g., the administration of booster vaccines). In addition to the dispersal of vaccines, apps can be used to monitor post-vaccination side effects. After finalizing the vaccination program, a digital immunity certificate or vaccination passport can be issued, which is always presentable by vaccinees [111]. This final key service of digital health is online issued evidence, which allows vaccinees to travel and access public places without restrictions. Recent evaluations showed that the digital vaccine passport had a positive impact on re-opening economies [112]. Importantly, the usage of mobile applications during the COVID-19 pandemic was confronted with some restrictions that originated from cultural, demographic and political issues in some country [107,113,114]. For instance, using mHealth for contact tracing and the registration of vaccine passports is highly controversial in some regions [115]. 

The COVID-19 pandemic has not only accelerated the digitalization of the health sector but also added new concepts by the introduction of digital vaccination passports, a series of nexuses such as the “Diplomacy for digital health,” “Digital health for diplomacy,” and “Digital health in diplomacy”. These developments have become more widespread on a global scale [114,116,117,118,119]. Undoubtedly, the COVID-19 pandemic was an important scene for the development and implementation of digital marketing tools, including social media and mobile apps, for health-related issues at a global level. These tools assisted health policymakers to implement a series of efficient responses to pandemic and will certainly be applied in pandemic preparedness programs. 

### 2.3. Engagement of Influencers in Public Health Issues

For market managers, the power of influencers is worthy [120]. Individuals with a large number of followers on social media can efficiently increase the selling rates of a products [121]. It has been shown that people like to follow individuals that have created their own follower community and are directly accessible and responsive to the audience. This contrasts with traditional celebrities that have a mass audience and are not tangible for ordinary people. Influencers, or micro-influencers, are trusted by the audience, and compared to known celebrities, their recommendations and advocations are often accepted by their followers [122,123]. Accordingly, influencers have an impact on decisions makers and can turn/drive the decisions of followers towards some specific products or opinions. There is an increasing body of evidence that social media influencers have a positive impact on public health issues [124,125] such as reducing smoking [126,127] and vaccination against HPV or influenza [95,128,129]. The use of influencers in public health is an intervention that can transmit tailored messages and inspire social media users to change their behaviors [95]. Furthermore, influencers can collaborate with public health policy makers and support them in the use of social media. Additionally, influencers’ population-targeted engagement has a positive impact on the health improvement of high-risk populations, e.g., immigrants [130,131,132,133,134]. 

It seems that all mentioned worthy experiences and knowledge from different disciplines in digital marketing have created a paved path for supporting health programs such as the elimination of widespread pathogens.

## 3. Viral Hepatitis Elimination and Applicable Digital Marketing

The eradication of infectious diseases requires a series of health measurements. If these tools are efficiently and correctly put in place, the infection of a targeted pathogen will be consequently controlled and eliminated (Figure 1) [135]. For instance, regulations that aim at the implementation of effective screening and identification of infected individuals provide a detailed picture of the epidemic. This information can be used for the distribution and administration of medicines and immunization efforts. The goal is to reduce the burden of disease and achieve long-lasting protection, which is crucial for a successful elimination program. However, in all elimination programs implemented by the WHO, vaccination plays a key role. Vaccination has the ability to block the transmission of pathogens and avoid new infections. For instance, mass vaccination supported the eradication program of smallpox, which succeeded in 1980, fourteen years after infections began in 1966 [136]. In addition to smallpox vaccination, the rapid development and implementation of vaccination strategies encourage public health policy makers to launch elimination programs for several pathogens such as rubella, measles and HPV [137,138,139].

### 3.1. Burden of Viral Hepatitis

Viral hepatitis is a global public health problem in which several viruses including hepatitis A (HAV), hepatitis B (HBV), hepatitis C (HCV), hepatitis D (HDV) and hepatitis E (HEV) are major causes of the infection and inflammation of liver tissue [140]. Considering its worldwide distribution, routes of transmission, virological characteristics, natural history and clinical outcomes, the global burden of viral hepatitis is mostly dedicated to HBV and HCV [141,142,143]. Globally, 58 million people live with chronic HCV and 296 million with HBV [140]. Furthermore, 90% of the annual global death rate of viral hepatitis (1.4 million) is related to HBV and HCV [144]. Geographical regions with limited access to safe water and sanitation also suffer from waterborne viral hepatitis agents (HAV and HEV) [145]. 

A broad spectrum of diagnostic platforms for HBV and HCV, including serological, molecular and biotechnological assays, are available [146,147,148]. Furthermore, several approaches can support diagnostic assays when different geographical distributions of viral genotypes or subgenotypes have an impact on diagnostic assays and clinical findings [149,150,151,152,153] or when different strains exert different responses to therapeutic measures [154,155]. Additionally, to stop viral hepatitis outbreaks, different investigation strategies have been proposed [156,157]. All these facilities have paved a path for an elimination program.

In 2016, the WHO adopted a global program to eliminate viral hepatitis by 2030 [158]. This program aims to decrease the incidence of chronic viral hepatitis infection and the mortality rates of infected patients by improving access to screening, universal vaccination and therapies [159]. This program was implemented when an efficient antiviral HCV therapy entered the market. These medications supported other available tools such as HBV vaccines and diagnostic assays. Different countries with variable levels of income and available facilities have tried to join this elimination program [160,161]. 

Because of the different prevalence of HBV and HCV, the success of the viral hepatitis elimination program relies on the application of tailored policies based on geographical region requirements [162]. In the implementation of a strong surveillance system in order to screen donated blood, pregnant women and persons with high-risk behavior are critical policies. This facilitates a series of effective prophylactic measures that can suppress the viral transmission in human communities [163]. In the case of HBV, the administration of vaccines and hepatitis B immune globulin (HBIG) to infants born to HBsAg-positive mothers is an important and time-dependent measurement [164]. In countries that included HBV vaccination in the national vaccination schedule, the prevalence of HBV has dramatically decreased [144]. Further, depending on the available infrastructures, different interventions have been applied to combat HCV [165]. The implementation of innovative strategy named “micro-elimination” has assisted policy makers in accelerating tackling HCV [166]. In micro-elimination, national elimination goals target subpopulations, and tailored services including treatment and prevention are quickly delivered to these groups. These populations are determined by some epidemiological factors, such as HIV/HCV coinfection, the need for blood transfusion (e.g., thalassemia, hemophilia, hemodialysis patients), prisoners, organ transplant recipients, people who inject drugs (PWID), children of HCV-infected mothers and immigrants originating from high-HCV-prevalence countries. Countries that apply micro-elimination are trying to achieve their goals within the WHO timeline [167]. Adding digital health technologies to these applied strategies can assist health authorities in monitoring and analyzing indicators and eventually filling in the existing gaps in this program. Compared to the other health issues, particularly infectious disease with a lack of vaccines (e.g., HIV), human pathogens with non-human hosts (e.g., Rabis) or infectious agents with a complicated control of their transmission (e.g., Influenza or Coronaviruses), viral hepatitis is well-known, and relying on advanced medical tools is controllable. Regarding this background, using digital tools can accelerate and catalyze measurements in line with an elimination program.

### 3.2. Current Status of Viral Hepatitis and Digital Technologies

Recent studies suggest that countries that use digital technology in their public health sectors are more active and successful in applying screening tests, disease monitoring and surveillance. These digital technologies provide platforms to connect public health sectors and politicians for the communication and dispersal of relevant information [168,169]. There are several ongoing global health programs, such as the viral hepatitis elimination issued by the WHO, that can benefit from the use of digital technologies. Many of these technologies have been used in the battle against COVID-19 but not yet for the viral hepatitis elimination program [170,171]. There are significant differences between these two infectious diseases, which also require different approaches in the use of digital health strategies. In contrast to COVID-19, viral hepatitis is a health problem with etiologies that have already evolved into different mounted genotypes and subgenotypes with particular geographical distribution [147,172]. Furthermore, there is a highly efficient and safe vaccine against HBV infection available, which is able to stop virus transmission. Although vaccinated individuals are well-protected against severe disease and death, the currently available SARS-CoV-2 vaccines only have a limited effect on transmission [173,174]. HBV vaccines provide long-term protection after three doses, while recent studies demonstrate that COVID-19 antibodies gradually wane after vaccination [174,175]. The waning of SARS-CoV-2 antibodies encouraged governments to offer booster vaccines to the population. In the case of HCV, no vaccine is currently available, although the recent antiviral medication is highly effective [142,176]. For both HBV and HCV, effective treatment and/or prevention strategies are available, and their proper implementation will significantly contribute to the success of the WHO elimination program. 

There is a large gap in the public perception of viral hepatitis, which needs an increase in public awareness and interest. Raising public awareness of viral hepatitis is a key step of the elimination program (Figure 1). In this regard, digital marketing platforms such as social media can play a significant role. Through different platforms of social media, knowledge about different types of hepatitis, modes of transmission viruses, clinical outcomes, diagnostic tests, available vaccines, prophylaxis and treatment can be promoted [177]. For instance, based on social media resources, researchers found that among Moroccan university students, knowledge about viral hepatitis and its routes of transmission and immunization is limited [178]. This gap of knowledge was also recognized by policy makers. In response, an “action plan for the health sector response to viral hepatitis in the WHO European Region” was developed in 2017. This plan included several targets that should be reached by 2020. One of the targets was to raise awareness in 50% of the people living with HBV and HCV. It is not surprising that in parts of the European continent, screening has not yet been well-implemented, and the majority of residents are not aware about their probable infections. This gap in public knowledge is much larger in other continents such as Africa and leads to the propagation of infection [144]. Additionally, in China, where the prevalence of HCV is high, many people are not aware of their infection [179]. The European action plan included five strategic directions and priority actions, and all highlighted the use of novel digital technologies [180]. 

The implementation of social media campaigns on Facebook and the communication of messages about HBV vaccination as a liver cancer prophylactic tool has been successful in elevating the knowledge of health providers and residences in Idaho, US [181]. Additionally, the application of social media to pinpoint existing hurdles that confront the elimination program has been fruitful. In some trials, these digital marketing tools have been able to supplement the knowledge of the general population about liver cancer and its screening. Despite the successful use of digital marketing tools to increase public knowledge, these campaigns did not improve their screening behavior [182]. Health-related messages on social media are mostly considered by young adults who originate from different regions [183]. For instance, social-media-based interventions promoted HBV screening among Korean residents in the US [131]. Social networking and mobile apps played an outstanding role in response to two massive Hepatitis A outbreaks in Europe and Canada. Of note, social media assisted health authorities in tracing the infected cases and implement a vaccination campaign to immunize susceptible individuals [184,185]. Further, social media has been used to inform people who inject drugs (PWID), heterosexual young people and MSM about viral hepatitis and to achieve a reduction in viral infection [186,187,188]. Furthermore, these platforms have also been used to update the knowledge of family physicians [189]. A survey reported a number of YouTube videos about HCV, which have attracted considerable viewers. However, these videos do not cover all aspects of HCV [190]. Data retrieved from social media about marginal communities can be very useful. For example, a survey in the US showed that awareness about hepatitis C virus is very low in a significant number of Hispanic and NH Asian young adults, who had never heard about HCV [191]. Authors of this study emphasized a social media campaign to educate targeted population about HCV screening. In Australia, low knowledge about viral hepatitis and its sequels in non-Australian-borne citizens is a serious issue [192]. Certainly, providing cultural- or ethnical-tailored health services through social media communications can be efficient. Subsequently, raising awareness about viral hepatitis improved screen and vaccination in the target population [193]. 

According to the WHO, vaccine hesitancy is one of the top threats for human life [194]. Social media is a frequently used platform to spread antivaccination messages and, therefore, is associated with the increase in vaccine hesitancy [195]. The effect of antivaccination messages on social media seems lower for pediatric HBV and HAV vaccinations compared to other vaccines such as for MMR and Rota virus. [195]. An analysis of tweets on the vaccine debate posted between 2006 and 2015 demonstrated that national vaccination programs affected messages on Twitter. It was deduced that vaccination programs can be improved by the application of social media campaigns. Twitter posts that included links to scientific websites were in accordance with the “*cross-platform use*” strategy [166,176,196]. 

### 3.3. Social Media and Data Mining in Viral Hepatitis Elimination

The implementation of healthcare programs requires access to and the meticulous analysis of relevant data. Technology giants such as Google and Apple have been deeply involved in collecting data related to smart digital devices. Mining and analyzing accurate information of digital communications provides valuable information regarding the preferences and daily behavior of people. For instance, the data generated through Google searches, Twitter and Wikipedia have been used for influenza surveillance. Through Bayesian Change Point Analysis, web-based search data from Influenza like illness (ILI) was compared with Centers for Disease Control and Prevention (CDC) ILI data. Interestingly, data from web searches were correlated with CDC data, which are routinely reported by health care providers in real time [197]. Similarly, social media data have been used to estimate viral hepatitis burden as well [198]. By means of social media analytic software, Symplur Signals (Symplur LLC), Twitter activity related to three chronic liver diseases (CLDs) including nonalcoholic fatty liver disease (NAFLD)/nonalcoholic steatohepatitis (NASH), hepatitis B and hepatitis C were analyzed between 2013 and 2019. This analysis showed a fluctuation in Twitter activity and tweeter impression during this period. However, given the trend of increasing Twitter activity for HBV, surpassing HCV by 2023 and 2024 is predicted [199]. Accordingly, through social media communication, the health status of people living with hepatitis could be traced and monitored in different geographical regions [200].

The use of smartphone apps and social media for viral hepatitis elimination results in the generation of big data. Analyzing these data with different platforms of artificial intelligence such as machine learning could support health authorities [195]. These analyses help policy makers implement surveillance through monitoring the migration of viral hepatitis reservoirs. The displacement of individuals [201] from high to low endemic regions [202] can be a major health concern for decision makers [160]. For instance, health care policies in line with a viral hepatitis elimination program are primarily based on country-specific demographic data, which can be challenged by the movement of people from various endemic regions [160,203]. 

### 3.4. Merging Strategies

A campaign is an effective method to communicate to a targeted population for a specific goal and at a specific time [204,205]. This strategy can be implemented in different geographical regions at local, national and international levels. Campaigns are cost-effective in distributing information and marketing to audiences [206]. Creating campaigns can also be used as an effective tool in conveying visions, missions, policies, marketing, public relations and health awareness or health education [207]. 

Effective health communications have an influence on the behavior of the targeted community. For experts in the health sector, communication is a vital step in a health program to offer prophylactic measures and improve the quality of life. Among different communication strategies, health- or disease-targeted campaigns are most popular. In fact, health campaigns can educate the targeted audience and improve health by changing their behavior. A campaign to promote breast feeding using breast milk (which is healthier for mother and child) is an example of such a health campaign [205]. Furthermore, promoting sports, combating obesity, stopping smoking and alcohol consumption, cancer screening, the use of vaccines and promoting sugar-free and diet drinks, low-fat milk, natural juices and lower caffeine consumption are other examples of public campaigns that successfully influenced their audience. [208,209,210].

In addition to other international health programs targeted at healthy lifestyle modifications, such as AIDS and sexually transmitted diseases, the viral hepatitis elimination program is suitable to be communicated through campaigns [211]. So far, campaigns for raising awareness about liver cancer, liver cancer screening and vaccination against hepatitis B and A have been implemented. However, there are some tips and critical points that could contribute to the success of health campaigns in this field [184,185,212]. 

Still, traditional media such as TV, radio and newspapers assist the health sector in creating and spreading health campaigns. They have proven to be successful in adjusting public perception and receiving considerable positive responses. Mass media (radio and TV) exposure has displayed a positive impact on childhood stunting in African countries [213]. Furthermore, vegetable and fruit consumption has received much attention after a campaign on media in the US and led to a positive change in behaviors [214]. The role of newspapers in a campaign for HPV vaccination in Japan [215] or the positive role of media to promote meningococcal vaccination in a recent outbreak in the Netherlands [216] corroborates the usefulness of traditional media. However, adding digital marketing, particularly social media, to this communication toolbox has massively reinforced health campaigns. It has been shown that the use of social media and the launch of public health promotion campaigns have great potential for the target audience [217]. Compared to other methods, launching campaigns on different platforms of social media amplifies bi-directional communication and efficiently leads to changes in the user’s behavior [124,218,219]. 

Elements such as availability, level of exposure to social media in targeted geographical region, appointing effective communication channels, presenting appropriate slogans or messages or visualized contents such as pictures or videos, also having a regular schedule, the determination of a target community and considering the educational level, age and gender of audiences [220,221,222,223] play critical role in a fruitful viral hepatitis campaign. In addition, one of the most important elements in campaigns is the choice of an appropriate message (warning, fear, excitement and pleasure) in line with the campaign goals. For example, messages containing fear are considered successful examples in antismoking campaigns. [224,225,226]. 

It has been shown that the use of innovative methods can increase the coverage of the message to the audience [227]. For example, in successful health campaigns, along with the use of images or videos, creative methods in storytelling and excitement have been emphasized [82,228,229]. Elements that contribute to the easy understanding of the message, such as infographics, had a positive impact on campaigns during the COVID-19 pandemic [223]. Furthermore, recruiting influencers, celebrities and seeders to the health campaign on social media is efficient and supports health campaigns [230,231]. Lessons learned from using influencers in campaigns that promoted flu vaccination demonstrated mass positive interaction [95,232] and could be applicable to other vaccination campaigns [82,233]. Influencers should be recruited from the health sectors. For example, in the case of a campaign for the elimination of viral hepatitis, inviting experts involved in prevention, screening and treatment is suitable for communicating the main message of campaigns [234,235]. Involving trained influencers such as family physicians, whose advice is listened to by the public, is always effective in the health sector. In a social network, they have a special position to recommend their patients to undertake viral hepatitis screening [236]. Providing a systematic and sustainable incorporation of messages about viral hepatitis by expert influencers fosters these campaigns. Educated influencers can present relevant messages and can easily convince their followers on social media through the messages they post [237]. Sufficient knowledge about relevant keywords and terms related to hepatitis screening, treatment, prevention and vaccination are principal necessities. Additionally, being familiar with key terms involved in viral hepatitis, such as antibodies, antigens or molecular biology terms such as PCR, Q-PCR and NGS and the usage of fibro scan or biopsy should be mastered by attracted influencers. This has been demonstrated previously, when hiring content knowledgeable influencers had a positive impact. These types of campaigns can reach large audiences, promote communications [24,238] and will finally decrease the burden of hepatitis in the short, middle and long term.

Finally, continuity and evaluation in dynamic communication are important for the promotion and development of a campaign [227]. A successful health campaign is updated by the received feedback [239,240,241]. In an evaluation system, the degree of change in the attitudes and behaviors of audiences who are exposed to the health campaign should be formulated and considered according to environmental and social factors [242,243]. The data obtained from the campaigns (psychological characteristics, demographics, along with the analysis of audience behavior on social media) should be used as effective tools in evaluating the campaign and determining future decisions [95,230].

In conclusion, digital technologies have revolutionized many disciplines and provided new opportunities for running healthcare programs, including viral hepatitis elimination. In this review, we tried to provide relevant evidence that the application of digital marketing tools such as mobile apps or creating campaigns on social media armed/engaged with influencers highly supports this program. The application of different strategies in using these tools will positively elevate public perception and some targeted populations. Subsequently, steps towards prophylactic screening and receiving appropriate treatment will be taken by the audience. Interactive communications between professional influencers and the community will neutralize the flood of vaccine hesitancy fueled by antivaccine movements. The success of these digital health strategies is closely insured by data monitoring, data mining and a continues evaluation system. Already, the application of social media has successfully been implemented in worldwide programs to control the spread of COVID-19. However, using digital marketing to promote the highly efficient vaccine for HBV and effective antiviral treatment for HCV is closer to the goal of viral hepatitis elimination than that of SARS-CoV-2. This should encourage health authorities to use digital platforms to decrease the worldwide burden of viral hepatitis and promote the viral hepatitis elimination program.

## Figures and Tables

**Figure 1 pathogens-11-00626-f001:**
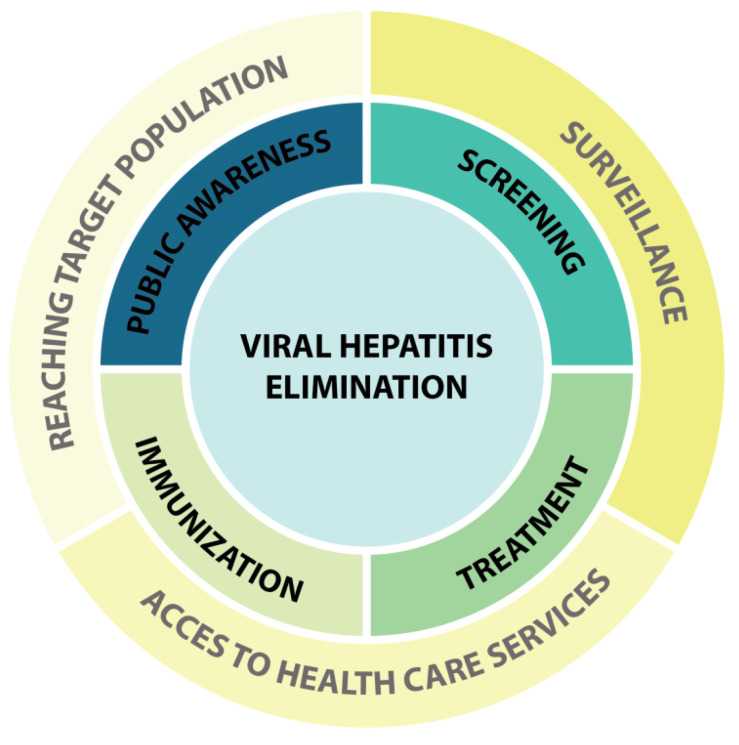
Application of digital marketing in program of viral hepatitis elimination. Center of diagram shows the goal of program. This goal is achieved by measurements such as increases in public awareness, screening, treatment and immunization against viral hepatitis (middle zoon). Medical apps, social media influencers and campaigns support and facilitate those medical services through surveillance, reaching target population and access to healthcare services (outer zoon).

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
