# Peer review of "Digital Marketing: A Unique Multidisciplinary Approach towards the Elimination of Viral Hepatitis"

_pathogens, 2022, doi:10.3390/pathogens11060626_

Round 1

Reviewer 1 Report

Dear Authors

After reviewing your article I have the doubt that it has been defined as a review but you have not shown any section of methodology that allows following the structure of the search that you have done.

It has been impossible for me to find this information and I think it is necessary that you provide it. And even if the authors have developed a narrative type of review, it should be indicated so that readers know the level of evidence provided by this type of document.

Author Response

Point-to-point response to reviewer

Response to Reviewer 1 Comments

After reviewing your article I have the doubt that it has been defined as a review but you have not shown any section of methodology that allows following the structure of the search that you have done.

It has been impossible for me to find this information and I think it is necessary that you provide it. And even if the authors have developed a narrative type of review, it should be indicated so that readers know the level of evidence provided by this type of document.

Response: Thanks for your comment and concern. We agree with the reviewer that this manuscript can be considered a narrative review. According to the suggestion of the reviewer and to clarify it for readers, we emphasized it in the abstract and introduction of the manuscript. Normally, a narrative review does not need a methodology section and is formed on the authors’ knowledge and expertise in additional to the relevant literature. The co-authors that collaborated in this work are experts in the field of viral hepatitis and social media management. Furthermore, scientific articles have been evaluated to support each section of this review. By relying on their own knowledge and expertise, co-authors are addressing a multidisciplinary approach to tackle a serious public health problem.

Reviewer 2 Report

My comments regarding this manuscript are:

  1. The abstract is good. But the abstract needs to add more sentences to focus on the topic of the digit technologies in the field of elimination of hepatitis
  2.  the introduction is well written.  I think two paragraphs need to be added at the end of the introduction. In the first paragraph, explain what is the contribution of the paper. In the second paragraph, describe other sections of the paper.
  3. At the beginning of the third part, add a paragraph to describe the specificity of the application of digit technologies in the elimination of hepatitis compared with other health issues.

Author Response

Point-to-point response to reviewer

Response to Reviewer 2 Comments

 1. The abstract is good. But the abstract needs to add more sentences to focus on the topic of the digit technologies in the field of elimination of hepatitis

 Response 1: We thank the reviewer for this suggestion which improves the abstract of the manuscript. Accordingly, we have added a couple of sentences to the abstract:

The presented evidence in our narrative review demonstrates that the application of digital marketing tools to create campaigns on social media, armed with professional influencers can efficiently consolidate this program. The application of different strategies in using these popular tools will raise the public awareness about viral hepatitis. Subsequently, the availability of an effective vaccine for HBV and antiviral medication for HCV can motivate the audience to take steps towards prophylaxis and screening methods against these infectious illnesses. The encouragement of health policy makers to apply digital communication technologies and comprehensive roadmaps to implement this global program will certainly decrease the burden of viral hepatitis worldwide.

2. the introduction is well written. I think two paragraphs need to be added at the end of the introduction. In the first paragraph, explain what is the contribution of the paper. In the second paragraph, describe other sections of the paper.

Response 2: Thanks for this constructive comment. We have added two paragraphs to the introduction.

First paragraph

In frame of a narrative review, we tried to dive into all available scientific evidence linked to the application of digital technologies in establishment of health programs. From both a technological and medical perspective, previous and current experiences helped us to evaluate the benefits, drawbacks, challenges, hurdles and gaps confronted with implementation of digital marketing for the viral hepatitis elimination program of WHO. Previous positive experiences with the use of digital technology on the management of health issues, particularly in control of vaccine preventable infectious diseases, highlight the message of this review. Here, we are addressing a series of appropriate strategies in the use of a digital toolbox to tackle a threatening public health issue. Based on our presented data, health authorities can accelerate action plans of viral hepatitis elimination and achieve the goals of the global of program.

Second Paragraph

The first chapter of this review evaluates a list of digital innovations and the impacts on human health through alterations in daily behavior and interests. In this part several examples demonstrate that data from communicative digital customized platforms have revolutionized health services (such as surveillance of pathogens or on time and appropriate actions) and stakeholders’ expectations.

In the following part of the text, we refer to the application of social media platforms and mobile apps during the COVID-19 pandemic. Furthermore, we shed light on the miss use of these digital tools by antivax movements. To build up the final part, we discuss the global burden of viral hepatitis and available medical tools for controlling this public health threat. We address digital tools that have been used for COVID-19 but are still missing in the viral hepatitis elimination program. Accordingly, to fill in these existing large gaps in public awareness and to use appropriately screening, vaccination and antivirals therapy against viral hepatitis, we urge health policy makers to apply digital marketing tools such as social media. Of note the use of these platforms and the recruitment of influencers is highly advised.

 At the beginning of the third part, add a paragraph to describe the specificity of the application of digit technologies in the elimination of hepatitis compared with other health issues.

 Response 3: Thanks for your comment. We added an appropriate paragraph to the third part.

Comparing to the other health issues, particularly infectious disease with lack of vaccines (e.g. HIV), human pathogens with non-human hosts (e.g. Rabies) or infectious agents with a complicated control of transmission (e.g. influenza or coronaviruses), viral hepatitis is a well-known pathogen. By relying on advanced medical tools, this pathogen is controllable by appropriate strategies of health authorities. Regarding this background, using digital tools can accelerate and catalyze measurements in line with the elimination program.

Round 2

Reviewer 2 Report

I have no more comments.